# Metabolic Syndrome Biomarkers of World Trade Center Airway Hyperreactivity: A 16-Year Prospective Cohort Study

**DOI:** 10.3390/ijerph16091486

**Published:** 2019-04-26

**Authors:** Sophia Kwon, George Crowley, Mena Mikhail, Rachel Lam, Emily Clementi, Rachel Zeig-Owens, Theresa M. Schwartz, Mengling Liu, David J. Prezant, Anna Nolan

**Affiliations:** 1Department of Medicine, Division of Pulmonary, Critical Care and Sleep Medicine, New York University, School of Medicine, New York, NY 10016, USA; Sophia.kwon@nyumc.org (S.K.); George.crowley@nyumc.org (G.C.); mena.mikhail@nyumc.org (M.M.); Rachel.Lam@nyumc.org (R.L.); Emily.clementi@nyumc.org (E.C.); 2Bureau of Health Services and Office of Medical Affairs, Fire Department of New York, Brooklyn, NY 11201, USA; Rachel.zeig-owens@fdny.nyc.gov (R.Z.-O.); Theresa.Schwartz@fdny.nyc.gov (T.M.S.); David.Prezant@fdny.nyc.gov (D.J.P.); 3Pulmonary Medicine Division, Department of Medicine, Montefiore Medical Center and Albert Einstein College of Medicine, Bronx, NY 10461, USA; 4Department of Epidemiology and Population Health, Albert Einstein College of Medicine, Bronx, NY 10461, USA; 5Department of Environmental Medicine, New York University, School of Medicine, New York, NY 10016, USA; Mengling.liu@nyumc.org; 6Division of Biostatistics, Departments of Population Health, New York University School of Medicine, New York, NY 10016, USA

**Keywords:** metabolic syndrome, airway hyperreactivity, World Trade Center

## Abstract

Airway hyperreactivity (AHR) related to environmental exposure is a significant public health risk worldwide. Similarly, metabolic syndrome (MetSyn), a risk factor for obstructive airway disease (OAD) and systemic inflammation, is a significant contributor to global adverse health. This prospective cohort study followed *N* = 7486 World Trade Center (WTC)-exposed male firefighters from 11 September 2001 (9/11) until 1 August 2017 and investigated *N* = 539 with newly developed AHR for clinical biomarkers of MetSyn and compared them to the non-AHR group. Male firefighters with normal lung function and no AHR pre-9/11 who had blood drawn from 9 September 2001–24 July 2002 were assessed. World Trade Center-Airway Hyperreactivity (WTC-AHR) was defined as either a positive bronchodilator response (BDR) or methacholine challenge test (MCT). The electronic medical record (EMR) was queried for their MetSyn characteristics (lipid profile, body mass index (BMI), glucose), and routine clinical biomarkers (such as complete blood counts). We modeled the association of MetSyn characteristics at the first post-9/11 exam with AHR. Those with AHR were significantly more likely to be older, have higher BMIs, have high intensity exposure, and have MetSyn. Smoking history was not associated with WTC-AHR. Those present on the morning of 9/11 had 224% increased risk of developing AHR, and those who arrived in the afternoon of 9/11 had a 75.9% increased risk. Having ≥3 MetSyn parameters increased the risk of WTC-AHR by 65.4%. Co-existing MetSyn and high WTC exposure are predictive of future AHR and suggest that systemic inflammation may be a contributor.

## 1. Introduction

Metabolic syndrome (MetSyn) is a clinical diagnosis made by fulfilment of at least three of the five following comorbidity criteria: Abdominal obesity, insulin resistance, hypertriglyceridemia, low high density lipoproteins (HDL), and hypertension [1,2]. MetSyn and particulate matter (PM) exposure are known independent risk factors in the development of many diseases including cardiovascular disease and cancer [3]. MetSyn, classically a risk factor for cardiovascular disease, is now being investigated as a risk factor for pulmonary disease [4].

Obesity, one component of MetSyn, has been typically linked to restrictive patterns of lung disease through mechanical stress and mass loading. However, many recent studies have focused on the systemic effects of MetSyn, through hormonal and immunoinflammatory mediators, and their association with pollution exposure and subsequent respiratory disease [5,6,7,8,9,10,11]. One study suggests that adipose tissue and adipokines such as C-reactive protein (CRP) and tumor necrosis factor-α (TNF-α) may contribute to a systemic low-grade inflammatory process leading to airway hyperreactivity (AHR) [12].

The association between MetSyn and the development of AHR has been seen in several studies [13]. Multiple cross-sectional studies have shown an increased prevalence of MetSyn or its constituents amongst those with diagnosed asthma or asthma-like symptoms [14,15,16]. A meta-analysis that included cohorts in the United States (US), Canada, and Europe reported that odds of incident asthma are increased by 50% in obese individuals, and that risk increased with body weight [17]. Two prospective studies investigated adults who were asthma-free at baseline and showed that obesity and insulin resistance were MetSyn risk factors that contributed to eventual asthma or asthma-like symptoms [17,18]. Murine studies showed that mice that developed insulin resistance from a high fat diet had increased airway resistance at baseline and after methacholine provocation, indicating a component of AHR [19].

AHR and PM exposure have also been strongly linked in numerous studies. In a cohort of asthmatic and non-asthmatic children exposed to freeway and non-freeway air pollution, there was a positive association between air pollution exposure and asthmatic children [20]. In a cohort study of 40 asthmatic children who attended school in close proximity to expressways, there was an increased risk of wheezing and shortness of breath [21]. In a cross-sectional study of adults over 50 years of age in low resource countries, 5.12% of cases were secondary to PM exposure, and the prevalence ratio of asthma after each 10 μg/m^3^ increase of PM_2.5_ was 1.05 [22]. The World Trade Center (WTC) complex destruction on 11 September 2001 (9/11) led to the release of over 11,000 tons of PM, and exposed over 300,000 local workers, residents, and rescue and recovery workers [23]. An early study monitoring pulmonary function in firefighters from the Fire Department of the City of New York (FDNY) with World Trade Center Particulate Matter (WTC-PM) exposure had AHR prevalence of 40%, and over half of the studied group had persistent AHR in a follow-up exam 10 years later [24,25]. These studies established a significant association between exposure level and AHR [26].

Our initial work focused on inflammatory biomarkers, such as GM-CSF and MDC, in WTC-PM-exposed firefighters [27]. We also investigated amylin, leptin, and lipids in a subset of exposed firefighters with WTC lung injury (WTC-LI) as defined by a loss of forced expiratory volume in 1 second (FEV_1_) to less than the lower limit of normal (LLN), and recently validated our findings of MetSyn associated with WTC-LI in the larger exposed group [4,28]. We now investigate the impact of MetSyn on the development of WTC-associated AHR.

## 2. Materials and Methods

Study Design: Demographics, clinical information and serial spirometry obtained as part of the Fire Department of New York World Trade Center Health Program (FDNY WTC-HP) was extracted from the FDNY electronic medical record [29]. All WTC-exposed FDNY rescue/recovery workers (baseline cohort; *N* = 12,781) were included if they were firefighters, had research consent, FEV_1_ ≥ lower limits of normal, no AHR on available lung function testing pre-9/11, fasting blood drawn prior to WTC site closure on 24 July 2002, and available clinical endpoints, yielding a source cohort of *N* = 7486 (Figure 1) [30]. Exposure to WTC-PM, defined per the FDNY WTC-HP, was based on first arrival at the WTC site and considered the highest if arrived in the morning of 9/11 during the collapse of the WTC, intermediate arriving the afternoon of 9/11, and lower intensity if arriving on or after 9/12 [31]. All subjects consented to analysis of their information for research at the time of enrolment. All data was collected in compliance with the Code of Federal Regulations, Title 21, Part 11 and the Montefiore Medical Center/Albert Einstein College of Medicine (#07-09-320) and New York University (#16-01412) Institutional Review Boards have approved this study. All participants gave informed written consent.

**AHR Definition.** The cohort was followed longitudinally until 1 August 2017 and *N* = 1906 had either a pulmonary function test (PFT) with assessment of bronchodilator response (BDR) or methacholine challenge test (MCT) administered. World Trade Center Airway Hyperreactivity (WTC-AHR) was defined at the earliest positive BDR or MCT after WTC exposure. In general, a bronchoprovocation test such as an MCT may be utilized to assess hyperreactivity, whereas a bronchodilator test may indicate reversibility consistent with asthma. MCTs were positive when the cumulative methacholine dose that reduced the FEV_1_ by 20%, (PC_20_) was equivalent or was less than 16 mg/mL [32]. BD was positive when post-bronchodilator FEV_1_ change exceeded 12% and at least 200 mL [33]. Those without AHR (*N* = 6947) were defined as those who either had a negative study or were presumed negative if they did not have subspecialty testing.

**MetSyn Phenotypic Definition.** Diagnosis of MetSyn was based on National Cholesterol Education Program Adult Treatment Panel III (NCEP ATP III) guidelines and optimized for our cohort by having at least 3 of 5 following criteria: Systolic blood pressure (SBP) ≥130 mmHg or diastolic blood pressure (DBP) ≥85 mmHg; HDL <40 mg/dL; triglycerides ≥150 mg/dL; insulin resistance, as glucose ≥100 mg/dL; or body mass index (BMI) ≥30 kg/m^2^. BMI ≥30 kg/m^2^ was used as surrogate for central adiposity as per Word Health Organization (WHO) guidelines [34]. Smoking information and exposure intensity were self-reported and collected from questionnaires administered during medical monitoring exams. Clinical parameters including DBP, glucose, and lipid panel were measured at WTC-HP entry. All of the above criteria were from data points obtained pre-9/11 and prior to any measurements of post-9/11 AHR.

**Additional Clinical Biomarkers.** We investigated the correlation between AHR and other clinical biomarkers. Absolute counts of differentiated white blood cells (WBC) such as neutrophils and eosinophils present at the first post-9/11 evaluation. A cutoff of at least 500 eosinophils was utilized, consistent with definition of clinical hyper-eosinophilia. Cholesterol/HDL ratio ≥3.5, a predictor of ischemic heart disease risk, and its association with AHR was also investigated [35,36,37].

**Statistical Analysis.** SPSS-23 (IBM) was used for primary data handling and statistics. Continuous variables are expressed as mean (standard deviation (SD)), and compared by two-sample t-test. Categorical data was summarized as count and proportions, and compared using Pearson-χ^2^. Smoking data was categorized into a dichotomous variable of ever or never a smoker, as previously described [38,39,40,41,42]. The primary endpoint of all analyses was development of WTC-AHR based on a positive BDR or MCT any time after 9/11. Survival interval was determined by time from 9/11 to positive AHR test or until 1 August 2017, the administrative censoring date of the study closure, if they did not have AHR. Association of endpoints and MetSyn, smoking, BMI, and exposure level were analyzed using the Cox proportional hazards regression and are represented as hazard ratio (HR) and 95% confidence interval (CI). We assigned a cut point of ≥500 eosinophils/µL as a marker of clinically significant eosinophilia for Cox modeling. All models were adjusted for age at 9/11, exposure intensity, and smoking status, and considered significant if *p* < 0.05. Omnibus testing was used to assess the quality of the comparisons. Time-to-event curves were determined by the Kaplan–Meier method and compared with the log-rank test. There were no dropouts in this study.

## 3. Results

### 3.1. WTC-AHR Cohort

Of the *N* = 1906 individuals with subspecialty pulmonary testing, *N* = 891 had a MCT while *N* = 1168 had BDR testing, and 153 individuals had both tests (Figure 1). Of the *N* = 539 participants that had WTC-AHR, 303 were positive on MCT only, 184 had a positive BDR only and 52 were positive on both tests, yielding a total of *N* = 355 who were MCT positive and *N* = 236 who were BDR positive (Figure 1). Demographic and clinical measures were compared between those with and without AHR (Table 1 and Table 2). Years of service information was only available on *N* = 5029/6947 non-AHR and *N* = 418/539 AHR. There was no difference between a positive MCT and a positive BDR in age, BMI, smoking status, or clinical measures except for total cholesterol. A positive MCT had a slightly higher cholesterol of mean (SD) of 217.4 (40.1) compared to positive BDR 210.2 (36.0), *p* = 0.03. AHR and non-AHR were not significantly different in race or smoking status; those with AHR were significantly older by a mean of 1 year, had an average of three fewer years of service compared to non-AHR, and had a higher percentage present in the morning of 9/11 (Table 1).

Spirometry was compared in those with AHR and non-AHR at both the most recent pre-9/11 and first post-9/11 examination showed significantly lower FEV_1_% predicted, Forced Vital Capacity (FVC) % predicted, and FEV_1_/FVC ratios in AHR, but the differences were clinically insignificant and were not indicative of OAD (Table 1).

Participants with a positive MCT had a PC_20_ mean (SD) of 5.6 (4.7) mg/mL. Participants with a positive BDR had a mean (SD) gain of FEV_1_ % predicted of 18.6% (9.9%), and 661.0 (274.0) mL. Participants with AHR also had a slightly higher average BMI (29.1 vs. 28.6 kg/m^2^; *p* = 0.001), triglycerides (*p* = 0.046), low-density lipoprotein (*p* = 0.028), and cholesterol (*p* = 0.009) compared to non-AHR (Table 2). A higher proportion of subjects with AHR also had MetSyn; subgroup analyses by Maentel–Haenszel odds ratio estimates shows that triglyceride ≥150 mg/dL (OR 21.6, 95% CI of 17.5–26.76, *p* < 0.001), obesity defined as BMI ≥30kg/m^2^ (OR 13.07, 95% CI 11.41–14.97, *p* < 0.001), and HDL <40 had OR of 12.25 (10.75–13.97), *p* < 0.001, were the most likely to meet the definition of MetSyn. SBP/DBP ≥130/85 had OR of 6.85 (6.03–7.79), *p* < 0.001, and glucose ≥ 100 OR 5.96 (5.24–6.78), *p* < 0.001.

Complete blood count and differentials were also compared on available samples between 532/537 AHR and 6896/6947 non-AHR (Table 2). AHR displayed higher WBC count and absolute eosinophil counts (AEC) compared to non-AHR. Of the *N* = 196 subjects with AEC > 500 cells/µL, *N* = 26 had AHR whereas *N* = 154 had at least one MetSyn risk factor.

### 3.2. Model Development

Cox proportional hazard models were used to estimate univariate hazard ratio of individual clinical biomarkers on developing WTC-AHR, with the adjustment of age, smoking, and WTC-exposure intensity (Table 3). AEC ≥500 cells/µL increased the risk of development of AHR by 94% (CI 1.31–2.88). The clinically utilized ratio of cholesterol/HDL ratio ≥3.5 was similarly correlated with a 33% increased risk of development of AHR. Similar to what was originally identified by t-test, dyslipidemia and obesity were significant risk factors for development of AHR, whereas hypertension or insulin resistance were not significant. This also mirrors the subgroup analyses where the most significant contributors to MetSyn definition were from dyslipidemia and obesity.

The final MetSyn model, adjusted for age, smoking, and exposure intensity, assessed the total number of MetSyn risk factors predicting AHR. Having at least two or three MetSyn risk factors had 69% and 65% increased risk of developing AHR, respectively (Table 3). Having high exposure, being present in the morning of 9/11, increased odds of developing AHR by 2.24 times, whereas being present in the afternoon increased odds by 1.76 times. Age also had an associated risk of development of AHR by 1.7% for every increasing year. Smoking was not significantly associated with AHR (Table 3).

Survival curves were plotted and time to divergence calculations from having no MetSyn risk factors showed that having one MetSyn risk factor diverged three years post-9/11, compared to having two risk factors, and divergence occurred within the first year post-9/11 (Figure 2). Kaplan–Meier curves were also assessed for subgroups of each MetSyn factors. BMI ≥30 kg/m^2^, triglycerides ≥150 mg/dL, HDL <40 mg/dL, and having the highest exposure to WTC-PM by being present at the site on the morning of 9/11 carried significantly higher risk of developing of AHR (Figure 3A–D).

We examined the reproducibility of the model in the cohort of *N* = 1906 who had MCT/BD testing. Even in this restricted cohort we found that having two or three MetSyn risk factors also yielded an increased risk of developing AHR by 54.4% and 39.6% (*p* = 0.001 and 0.014) respectively. Having one MetSyn risk factor had 25% increased risk but was not statistically significant (*p* = 0.08), and exposure was not a significant risk factor in the smaller cohort.

## 4. Discussion

The WTC-exposed FDNY rescue/recovery workers represent the largest longitudinally assessed first responder cohort with pre and post lung function assessments following a high PM exposure. They continue to have their health adversely impacted even after 18 years [26,27,28,43,44,45,46,47,48,49,50,51,52,53,54]. Our previous work focused on the contribution of MetSyn in the development of WTC-LI [4,28]. We now show that metabolically active biomarkers and markers of inflammation (such as eosinophils) predict AHR in the WTC-exposed firefighter cohort [55,56,57].

This study represents the only longitudinal study to our knowledge investigating the temporal relationship of MetSyn, PM exposure, and AHR. We demonstrate that MetSyn is an independent risk factor for the development of AHR, as all study participants were categorized as having or not having MetSyn prior to the development of AHR. A prior cross-sectional study investigating MetSyn, PM exposure, and cardiovascular risk found no significant associations with inflammatory markers of CRP or WBC [3].

Our earlier model showed that dyslipidemia and heart rate independently increased the odds of developing WTC-LI [4,28]. While WTC-LI and AHR both fall under the umbrella of OAD, there is little overlap in these populations. When examining a subgroup analysis of the WTC-LI cases, only *N* = 203/1204 (16.8%) also have AHR. This strongly suggests that MetSyn is implicated in multiple pathways in the development of OAD. Although MetSyn risk factors and cholesterol/HDL ratios are classically predictors of cardiovascular disease, their implications in affecting future lung disease are novel. Moreover, these represent reversible risk factors that may be potential therapeutic targets to alter the outcome of other obstructive lung disease.

Our current investigation shows the associations of MetSyn biomarkers with the development of WTC-AHR. We also show obesity having similar ability to predict AHR, an unexpected finding given the vast literature of restrictive patterns in obesity. However, this fits in the growing body of literature showing that obesity has hormonal pathways that influence the pulmonary environment. Another unexpected finding, but similar to our prior investigation of MetSyn predictor of WTC-LI, was that glucose was not a significant predictor of WTC-AHR [28]. Specifically, in our current study with cut-points of 100 and 126 mg/dL, there was no significantly increased risk of lung injury in our cohort. Studies in non-exposed individuals have shown an association between insulin resistance and OAD [58].

A strength of this study is the rigorously characterized prospective cohort, with a clearly defined time of exposure (9/11), pre and post exposure lung function measurements, blood drawn soon after exposure, and MetSyn categorization done prior to AHR measurements. Another strength of this study is that the development of AHR was post-exposure in that FDNY firefighters are excluded at hire and during annual medical monitoring if they have signs or symptoms of airways obstruction or AHR [59]. A key strength of this study is in the study design that allowed us to explore multiple aspects of WTC-AHR. Since the decision to administer BDR or MCT tests to determine disease is often a clinician’s judgement, it was reassuring to have found little difference between the two groups, those with only a positive BDR and those with only a positive MCT. This suggests that our findings are plausible and potentially reproducible in other cohorts. Examining the model for reproducibility in the smaller cohort of *N* = 1906 also bolsters our findings that MetSyn is an independent risk factor for development of future AHR.

Using the clinical markers of MetSyn as predictors of AHR is advantageous in several ways. These biomarkers are easily attainable, cost effective, and can be replicated in many cohorts. Dyslipidemia, insulin resistance, obesity, and hypertension are also all potentially reversible causes of end-organ disease. Extending statin therapy, for example, and increasing glycemic control can be the target of future studies focused on their mitigating effects in progressive lung disease. Furthermore, our group is currently studying dietary effects on pulmonary function and biomarker profile of the FDNY exposed cohort. MetSyn also has global reach and can be investigated in other cohorts with similar pollutant or PM exposure.

Eosinophilia, while a significant risk factor of AHR, did not augment the predictive model investigating MetSyn. Subgroup analysis shows that although only 13% of the hyper-eosinophilia group had overlap with the AHR population, 78.5% (*N* = 154/196) had at least one MetSyn risk factor. This was somewhat surprising because the literature reflects that obesity-related lung dysfunction is often a neutrophil-mediated process [60]. This may indicate the need for further studies in eosinophilia-mediated pathways in MetSyn.

There are several limitations to this study. We focus on a male-only cohort. Interestingly, MetSyn and asthma have been shown to have stronger associations in female-only cohorts [18,61]. An additional limitation is that we classified participants as not having AHR if they had negative tests or even if they had no testing of AHR. The presumption of negative AHR in the absence of direct measurement, while a limitation, biased our results towards the null. We also used a broad definition of AHR by requiring either a positive BDR or a positive MCT. This combined definition could have biased our results away from the null and towards a positive finding but we believe this combined definition is the most clinically relevant, and also accounts for those that may have only had one type of test due to clinical contraindications [32].

We also caution on over-interpretation of the results of individual components of MetSyn. We use a one-time fasting blood level at the earliest time point after exposure to 9/11 to make predictions of future lung injury. Future studies can focus on repeated measures of MetSyn, dyslipidemia, fasting blood glucose, and AHR to control for other possible confounders and aid in understanding the associations of individual components of MetSyn and AHR and their temporal relationships. The lack of medication history could also cause under-identification of those with MetSyn risk factors. Having this information could improve the sensitivity of those at risk of AHR and specificity of the model. Alternatively, it can help to determine if attenuation of MetSyn risk factors through pharmacologic methods impacted pulmonary health.

## 5. Conclusions

In summary, MetSyn biomarkers are predictors of future WTC-AHR in a large cohort of WTC-exposed FDNY firefighters followed over 16 years. These metabolically active biomarkers are associated with dyslipidemia, insulin resistance, and cardiovascular disease, and suggest that MetSyn may contribute to systemic inflammation that leads to future development of AHR. Our data supports the hypothesis and contributes to the growing body of literature investigating the complex associations between potentially reversible MetSyn risk factors and lung injury. We are strongly encouraged by our results indicating that pathways involved in metabolism have broad impacts on the immune and hormonal environment in the lung.

## Figures and Tables

**Figure 1 ijerph-16-01486-f001:**
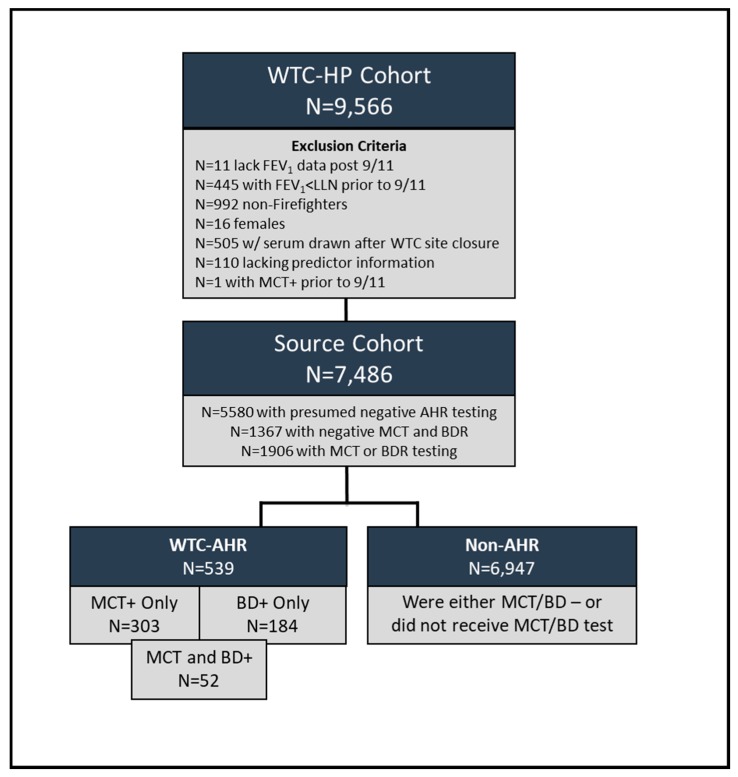
Study Design. Fire Department of New York (FDNY) rescue workers exposed to World Trade Center (WTC) particulates and enrolled in the WTC Health Program.

**Figure 2 ijerph-16-01486-f002:**
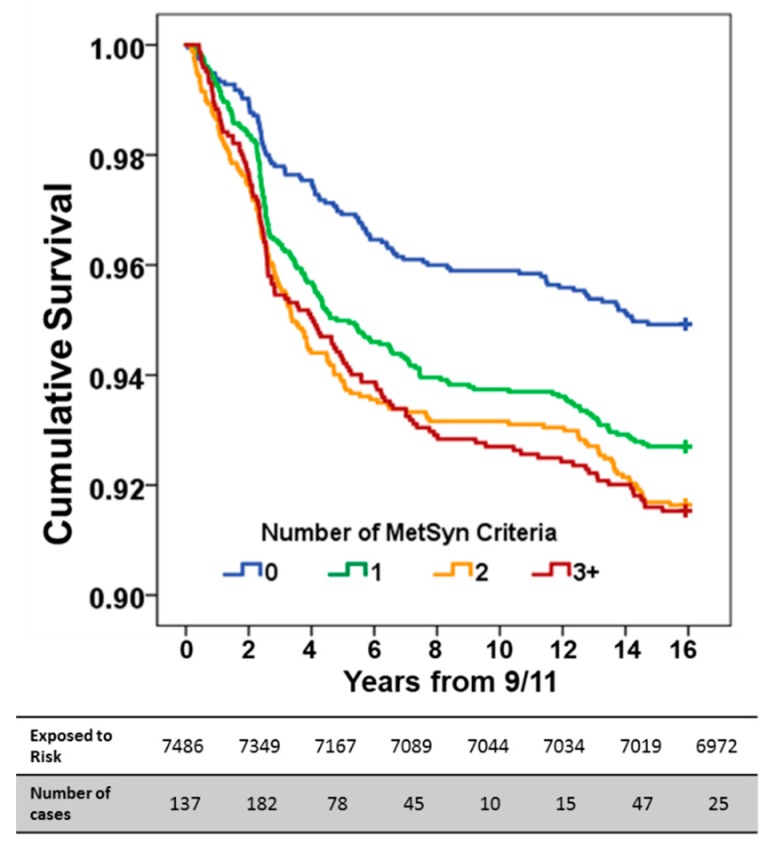
Cumulative AHR survival curves by total number of MetSyn biomarkers. Cumulative disease-free survival is expressed on the *y*-axis and time in years from their WTC exposure is on the *x*-axis. Life table expresses the number of individuals at risk in 2-year intervals.

**Figure 3 ijerph-16-01486-f003:**
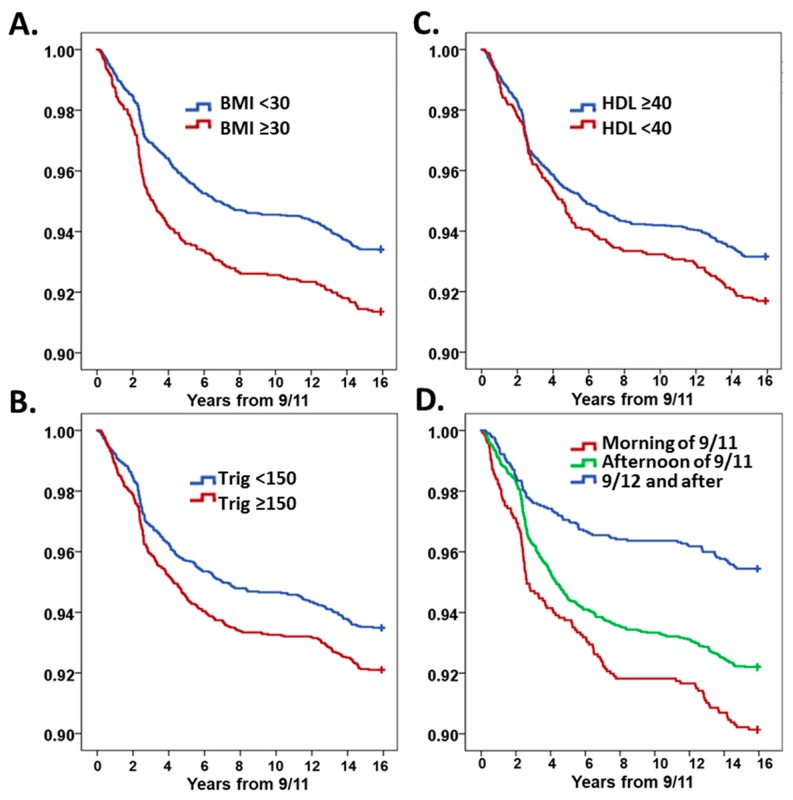
Kaplan–Meier survival curves stratified by (**A**). BMI ≥30 kg/m^2^ (*p* = 0.001 by log rank), (**B**). Triglycerides ≥150 mg/dL (*p* = 0.019), (**C**). HDL <40 mg/dL (*p* = 0.038), and (**D**). Exposure intensity (*p* < 0.001). Cumulative disease-free survival is expressed on the *y*-axis and time in years from their WTC exposure is on the *x*-axis. Log rank was not significant for SBP, DBP, and glucose, and were not included in this graph.

**Table 1 ijerph-16-01486-t001:** Demographic and pulmonary function test data of group.

Measure		MCT+ *N* = 355	BD+ *N* = 236	Non-AHR *N* = 6947	WTC-AHR *N* = 539	*p*
Age on 9/11		40.3 (6.3)	40.9 (6.8)	39.5 (7.5)	40.5 (6.5)	0.004
Years of service *		19.3 (7.3)	20.1 (7.2)	22.8 (6.5)	19.7 (7.3)	<0.001
Ever smokers		122 (34%)	89 (38%)	2531 (36%)	191 (35%)	0.643
Race	Caucasian	342 (96%)	220 (93%)	6529 (94%)	512 (95%)	0.560
	African American	3 (1%)	6 (3%)	183 (3%)	9 (2%)	
	Hispanic	9 (3%)	10 (4%)	215 (3%)	17 (3%)	
	Asian/other	1 (.3%)	0 (0%)	20 (0.3%)	1 (0.2%)	
Exposure group	Morning of 9/11	79 (22%)	59 (25%)	1124 (16%)	123 (23%)	<0.001
	Afternoon of 9/11	211 (59%)	134 (57%)	3749 (54%)	317 (59%)	
	On or after 9/12	65 (18%)	43 (18%)	2074 (30%)	99 (18%)	
Pre-9/11	FEV_1_% pred.	101.5 (11.6)	101.5 (13.5)	106.3 (13.0)	101.7 (12.4)	<0.001
	FVC% pred.	97.7 (10.9)	98.4 (12.7)	99.6 (12.1)	98.2 (11.7)	0.017
	Ratio	82.9 (5.7)	82.4 (5.4)	85.3 (4.9)	82.7 (5.6)	<0.001
WTC-HP entry	FEV% pred.	91.8 (13.4)	90.7 (14.8)	98.1 (13.1)	91.6 (13.9)	<0.001
	FVC% pred.	89.6 (12.0)	89.3 (21.1)	92.4 (11.8)	89.6 (12.2)	<0.001
	Ratio	82.0 (6.0)	81.2 (6.6)	84.6 (4.9)	81.7 (6.3)	<0.001

Values are in mean (SD) or *N* (%) as indicated. *p* calculated by *t*-test or Chi-square as appropriate, comparing airway hyperreactivity (AHR) and non-AHR. * Data available on *N* = 5029/6947 non-AHR, *N* = 418/539 AHR.

**Table 2 ijerph-16-01486-t002:** Clinical measures of inflammation and metabolic syndrome.

Measure	MCT+ *N* = 355	BD+ *N* = 236	Non-AHR *N* = 6947	WTC-AHR *N* = 539	*p*
Systolic BP, mmHg	118.0 (12.5)	118.3 (12.6)	117.1 (12.5)	118.0 (12.7)	0.092
Diastolic BP, mmHg	73.4 (8.0)	74.3 (8.3)	73.4 (8.4)	73.6 (8.2)	0.598
BMI at WTC-HP entry, kg/m^2^	29.2 (3.2)	28.9 (3.0)	28.6 (3.3)	29.1 (3.2)	0.001
White blood cells × 10^9^ cells/L *	6.5 (1.9)	6.5 (1.8)	6.3 (1.6)	6.5 (1.9)	0.021
Neutrophils (ANC)	3809.8 (1654.5)	3697.7 (1400.1)	3664.6 (1290.1)	3758.1 (1524.8)	0.113
Lymphocytes (ALC)	1818.9 (534.4)	1861.8 (590.2)	1830.9 (540.3)	1843.4 (561.2)	0.608
Eosinophils (AEC)	227.8 (149.9)	227.2 (160.3)	187.1 (130.9)	229.3 (156.6)	<0.001
Monocytes (AMC)	579.4 (194.3)	605.7 (216.2)	581.3 (193.2)	591.1 (204.8)	0.263
Glucose	92.9 (18.8)	91.7 (10.4)	91.6 (13.9)	92.5 (16.4)	0.177
Triglyceride	195.7 (139.1)	190.7 (126.0)	185.1 (136.6)	197.4 (137.9)	0.046
HDL	48.0 (12.6)	47.1 (12.1)	48.1 (11.7)	47.6 (12.4)	0.351
LDL	133.1 (34.5)	128.3 (32.4)	128.3 (33.5)	131.6 (33.7)	0.028
Cholesterol	217.4 (40.1)	210.2 (36.0)	210.8 (38.7)	216.3 (38.5)	0.009
Cholesterol/HDL ratio	4.8 (1.5)	4.7 (1.3)	4.6 (1.4)	4.8 (1.4)	0.007
MetSyn definition	82 (23%)	54 (23%)	1329 (19%)	123 (23%)	<0.001
SBP ≥ 130 and/or DBP ≥ 85 mmHg	78 (22%)	56 (24%)	1384 (20%)	119 (22%)	0.229
HDL < 40 mg/dL	94 (27%)	72 (31%)	1667 (24%)	151(28%)	0.036
Triglycerides ≥ 150 mg/dL	194 (55%)	123 (52%)	3428 (49%)	294 (55%)	0.020
Glucose ≥ 100 mg/dL	72 (20%)	45 (19%)	1269 (18%)	106 (20%)	0.419
BMI ≥ 30 kg/m^2^	130 (30%)	78 (33%)	2040 (29%)	193 (36%)	0.002

Values are in mean (SD) or *N* (%) as indicated; *p* calculated by t-test or Chi-square as appropriate, comparing AHR and non-AHR; * Data available on *N* = 6896/6947 non-AHR, *N* = 532/537 AHR, differentials expressed as absolute counts, cells/µL.

**Table 3 ijerph-16-01486-t003:** Cox proportional hazards of univariate metabolic risk factors of AHR.

Measure		Hazards (95% CI)
Cholesterol/HDL ratio ≥ 3.5		1.332 (1.057–1.679)
BMI ≥ 30 kg/m^2^		1.329 (1.114–1.585)
Glucose ≥ 100 mg/dL		1.062 (0.857–1.315)
Lipids mg/dL	HDL < 40	1.237 (1.025–1.492)
	Triglycerides ≥ 150	1.204 (1.016–1.427)
Blood pressure mmHg	Systolic ≥ 130	1.079 (0.872–1.335)
	Diastolic ≥ 85	0.970 (0.718–1.309)
Number of MetSyn risk factors	1	1.441 (1.124–1.847)
	2	1.690 (1.310–2.151)
	3+	1.654 (1.268–2.158)
Exposure intensity	Morning of 9/11	2.240 (1.719–2.919)
	Afternoon of 9/11	1.759 (1.403–2.205)
	After 9/12	Reference
Ever smoker		1.759 (1.403–2.205)
Age (per year)		1..017 (1.005–1.029)

All models were adjusted for age, smoking, and exposure intensity. Exposure, age, and smoking RR refer to RR in final model of combined MetSyn risk factors.

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
