# Peer review of "Metabolic Syndrome Biomarkers of World Trade Center Airway Hyperreactivity: A 16-Year Prospective Cohort Study"

_ijerph, 2019, doi:10.3390/ijerph16091486_

Round 1
Reviewer 1 Report
Dear authors,
A very interesting paper about the longterm effects of the disaster of WTC in 2001.OAD is not in the glossary, might be better.
What might be interesting is also to look for renal disease since there seems to be a systemic inflammation caused by Metsyn factors leading to lung injury. I wonder if the authors can give data on renal dysfunction in their cohort.
Author Response
Dear Reviewer,
Thank you for your critique.
Please find below a point by point response.
OAD is not in the glossary, might be better.
We have added OAD to the glossary and included the use of the abbreviation in the manuscript.
What might be interesting is also to look for renal disease since there seems to be a systemic inflammation caused by Metsyn factors leading to lung injury. I wonder if the authors can give data on renal dysfunction in their cohort.
We agree that this is a very interesting set of variables that need to be examined. Unfortunately, at this time we do not have a complete database with creatinine (and associated relevant variables) available.
We did examine creatinine values of a pilot group that had been used for of our prior biomarker studies and had creatinine values for 1099/1720.
As you can see in the following frequency table, most of the population (99.5%) had a normal renal function, assuming chronic kidney disease with a Cr of >1.5.
If we assume similar trends in the N=7486, and purport that perhaps at least at the first blood test post 9/11, renal dysfunction was not readily apparent or contributory to the picture of chronic inflammation leading to WTC-LI. We have left this information out of the main manuscript due to having incomplete information.

Reviewer 2 Report
In their manuscript “Metabolic Syndrome Biomarkers of World Trade Center Airway Hyperreactivity: A 16-Year Prospective Cohort Study” the authors investigated the association of clinical biomarkers of MetSyn with AHR development in a cohort exposed to the particular matter generated during the World Trade Center incident in 9/11/2001.
The thematic of research is well introduced and the results nicely discussed. However, I have some remarks considering the abstract formulation and results presentation.
Concerning the abstract, please improve the information on the cohort on which you work on as well as on the biomarkers that you took in consideration. For example, Line 21 you can mention that the cohort followed is composed of people exposed during the World Trade Center incident.
In the Results section, overall results cited in the text have to appear, if possible, in Tables. If there is not the case, please mention (data not shown). As results on the overall cohort and on the MCT/BD cohort are presented in the text, it will be nice to have at least one table with MCT/BD cohort data (N=1,906) in order to help the reader to follow what you mentioned in the text.
Table 1A:
Please explain in the text how do you obtained the N=355 MCT+ as well as N=236 BD+. As well as N=5029 and 418 in the mention “Data available on N=5029/6947 non-AHR, N=418/539 AHR.”
Lines 152 – “Table 1B” has to go in the previous line.
Lines152-153: please show the statistical results that support your affirmation “subgroup analyses shows that dyslipidemia and obesity defined as BMI ≥30kg/m2 were the most likely contributors”
Lines 154-155: please detailed what you mean by “N=26/532 had AHR”. Similarly for “N=154/196 had at least one MetSyn risk factor”
The results presented on Table 1A and 1B seems to not be obtained from the same population size:
Table 1A : Data available on N=5029/6947 non-AHR, N=418/539 AHR
Table 1B : Data available on N=6896/6947 non-AHR, N=532/537 AHR
Why? How it could be?
Lines 160-161: Why the information: “AEC≥500cells/μL increased the risk of development of AHR by 94% (CI 1.31-2.88).” doesn’t appear in the Table2?
Lines 163-164: Link this conclusion “dyslipidemia and obesity were significant risk factors in development of AHR whereas hypertension or insulin resistance were not significant” to the variables quantified and the data presented in Table2.
Lines 167-170: Data corresponding to those sentences have to appear somewhere in a Table.
General remark:
Please mention please the significance of the acronyms the first time that you cite them in the text, even if you list the acronyms at the end of the article.
- Line 24: WDR-AHR
- Line 68: WTC-PM
- Line 77: FDNY
- Line 78: FEV1
- Line 81: FDNY-WTCHP
- Line 89 : BDR and MCT
- Line 94 : BD
- Line 97 : NCEP ATP III
- Line 154: WBC
- Line 198 : WTC-LI
Add WDR the first time that you cite the World Trade Center (in the abstract and at the line 65 in the introduction)
BD and BDR were both used in the text. I will suggest to use only one of the two everywhere in the text.
Author Response
REVIEWER 2.
Concerning the abstract, please improve the information on the cohort on which you work on as well as on the biomarkers that you took in consideration. For example, Line 21 you can mention that the cohort followed is composed of people exposed during the World Trade Center incident.
The abstract now states on line 21:
“This prospective cohort study followed N=7,486 World Trade Center (WTC) exposed male firefighters from 9/11/2001 until 8/1/2017, and investigated N=539 with newly developed AHR for clinical biomarkers of MetSyn and compared them to the non-AHR group. Male firefighters with normal lung function and no AHR pre-9/11 that had blood drawn from 9/11/2001-7/24/2002 were assessed. World Trade Center-Airway Hyperreactivity (WTC-AHR) was defined as either a positive bronchodilator response (BDR) or methacholine challenge test (MCT). The electronic medical record (EMR) was queried for their MetSyn characteristics (lipid profile, body mass index (BMI), glucose), and routine clinical biomarkers (such as complete blood counts).”
We also state at the end of the introduction Line 79:
“Our initial work focused on inflammatory biomarkers in WTC-PM exposed firefighters such as GM-CSF, MDC.1 We also investigated amylin, leptin, and lipids in a subset of exposed firefighters with WTC-lung injury (WTC-LI) as defined by a loss of forced expiratory volume in 1 second (FEV1) to less than the lower limit of normal (LLN), and recently validated our findings of MetSyn associated wit WTC-LI in the larger exposed group. 2,3 We now investigate the impact of MetSyn on the development of WTC associated AHR.”
In the Results section, overall results cited in the text have to appear, if possible, in Tables. If there is not the case, please mention (data not shown). As results on the overall cohort and on the MCT/BD cohort are presented in the text, it will be nice to have at least one table with MCT/BD cohort data (N=1,906) in order to help the reader to follow what you mentioned in the text.
We have included the description of the 1906 in Figure 1. The authors have elected to not include their demographic data in the tables as the population is a mix of those with positive AHR/negative testing / mixed results that may not contribute any additional meaningful information, and may potentially confuse the readers further.
Table 1A: Please explain in the text how do you obtained the N=355 MCT+ as well as N=236 BD+. As well as N=5029 and 418 in the mention “Data available on N=5029/6947 non-AHR, N=418/539 AHR.”
Line 141 now states: Of the N=539 participants that had WTC-AHR, 303 were positive on MCT only, 184 had a positive BDR only and 52 positive on both tests, yielding a total of N=355 who were MCT positive, and N=236 who were BDR positive, Figure 1.
Line 145 now states: Years of service information was only available on N=5029/6947 non-AHR and N=418/539 AHR.
Lines 152 – “Table 1B” has to go in the previous line.
This has now been fixed and shows on line 162.
Lines152-153: please show the statistical results that support your affirmation “subgroup analyses shows that dyslipidemia and obesity defined as BMI ≥30kg/m2 were the most likely contributors”
We now state On line 163: subgroup analyses by Maentel-Haenszel Odds Ratio estimates shows that triglyceride≥150mg/dL (OR 21.6, 95%CI of 17.5-26.76, p<0.001), obesity defined as BMI ≥30kg/m2 (OR 13.07, 95%CI 11.41-14.97, p<0.001), and HDL<40 had OR of 12.25(10.75-13.97), p<0.001, were the most likely contributors to definition of MetSyn. SBP/DBP≥130/85 had OR of 6.85(6.03-7.79), p<0.001, and glucose≥100 OR 5.96(5.24-6.78), p<0.001.
Lines 154-155: please detailed what you mean by “N=26/532 had AHR”. Similarly for “N=154/196 had at least one MetSyn risk factor”
We clarify this on line 170: Of the N=196 subjects with AEC>500 cells/µL, N=26 had AHR whereas N=154 had at least one MetSyn risk factor.
The results presented on Table 1A and 1B seems to not be obtained from the same population size:
Table 1A: Data available on N=5029/6947 non-AHR, N=418/539 AHR
Table 1B : Data available on N=6896/6947 non-AHR, N=532/537 AHR
Why? How it could be?
This was referring to an asterisked component of the tables, Years of Service in Table 1A, and WBC in Table 1B. This has been clarified in line 145: Years of service information was only available on N=5029/6947 non-AHR and N=418/539 AHR.
And Line 169: Complete blood count and differentials were also compared on available samples between 532/537 AHR and 6896/6947 non-AHR, Table 1B
Lines 160-161: Why the information: “AEC≥500cells/μL increased the risk of development of AHR by 94% (CI 1.31-2.88).” doesn’t appear in the Table2?
Table 2 is focused on Metabolic Risk Factors alone. We wanted to focus this analysis on Metabolic Syndrome biomarkers, and the CBC data was not complete on all individuals as explained above.
Lines 163-164: Link this conclusion “dyslipidemia and obesity were significant risk factors in development of AHR whereas hypertension or insulin resistance were not significant” to the variables quantified and the data presented in Table2.
Line 179 now reads: Similar to what was originally identified by t-test, dyslipidemia and obesity were significant risk factors in development of AHR whereas hypertension or insulin resistance were not significant. This also mirrors the subgroup analyses where the most significant contributors to MetSyn definition were from dyslipidemia and obesity.
Lines 167-170: Data corresponding to those sentences have to appear somewhere in a Table. We have now included this information in Table 2
Please mention please the significance of the acronyms the first time that you cite them in the text, even if you list the acronyms at the end of the article.
- Line 24: WTC-AHR
- Line 68: WTC-PM
- Line 77: FDNY
- Line 78: FEV1
- Line 81: FDNY-WTCHP
- Line 89: BDR and MCT
- Line 94: BD
- Line 97: NCEP ATP III
- Line 154: WBC
- Line 198: WTC-LI
Add WDR the first time that you cite the World Trade Center (in the abstract and at the line 65 in the introduction)
We have now uniformly edited the manuscript to include an explanation of the acronym when it is 1st mentioned and in the abbreviations list.
BD and BDR were both used in the text. I will suggest to use only one of the two everywhere in the text.
Agreed, and we have now uniformly used BDR throughout the manuscript.

Round 2
Reviewer 2 Report
A nice work was done to improve the manuscrit.